# Variational Dynamic Mixtures

## Abstract

Deep probabilistic time series forecasting models have become an integral part of machine learning. While several powerful generative models have been proposed, we provide evidence that their associated inference models are oftentimes too limited and cause the generative model to predict mode-averaged dynamics. Mode-averaging is problematic since many real-world sequences are highly multi-modal, and their averaged dynamics are unphysical (e.g., predicted taxi trajectories might run through buildings on the street map). To better capture multi-modality, we develop variational dynamic mixtures (VDM): a new variational family to infer sequential latent variables. The VDM approximate posterior at each time step is a mixture density network, whose parameters come from propagating multiple samples through a recurrent architecture. This results in an expressive multi-modal posterior approximation. In an empirical study, we show that VDM outperforms competing approaches on highly multi-modal datasets from different domains.

## 1 Introduction

Making sense of time series data is an important challenge in various domains, including ML for climate change. One important milestone to reach the climate goals is to significantly reduce the $CO_2$ emissions from mobility (Rogelj et al., 2016). Accurate forecasting models of typical driving behavior and of typical pollution levels over time can help both lawmakers and automotive engineers to develop solutions for cleaner mobility. In these applications, no accurate physical model of the entire dynamic system is known or available. Instead, data-driven models, specifically deep probabilistic time series models, can be used to solve the necessary tasks including forecasting.

The dynamics in such data can be highly multi-modal. At any given part of the observed sequence, there might be multiple distinct continuations of the data that are plausible, but the average of these behaviors is unlikely, or even physically impossible. Consider for example a dataset of taxi trajectories[1]. In each row of Fig. 1a, we have selected 50 routes from the dataset with similar starting behavior (blue). Even though these routes are quite similar to each other in the first 10 way points, the continuations of the trajectories (red) can exhibit quite distinct behaviors and lead to points on any far edge of the map. The trajectories follow a few main traffic arteries, these could be considered the main modes of the data distribution. We would like to learn a generative model of the data, that based on some initial way points, can forecast plausible continuations for the trajectories.

Many existing methods make restricting modeling assumptions such as Gaussianity to make learning tractable and efficient. But trying to capture the dynamics through unimodal distributions can lead either to "over-generalization", (i.e. putting probability mass in spurious regions) or on focusing only on the dominant mode and thereby neglecting important structure of the data. Even neural approaches, with very flexible generative models can fail to fully capture this multi-modality because their capacity is often limited through the assumptions of their *inference model*. To address this, we develop variational dynamic mixtures (VDM). Its generative process is a sequential latent variable model. The main novelty is a new multi-modal variational family which makes learning and inference multi-modal yet tractable. In summary, our contributions are

- **A new inference model.** We establish a new type of variational family for variational inference of sequential latent variables. By successively marginalizing over previous latent states, the procedure can be efficiently carried-out in a single forward pass and induces a multi-modal posterior

---

[1] https://www.kaggle.com/crailtap/taxi-trajectory

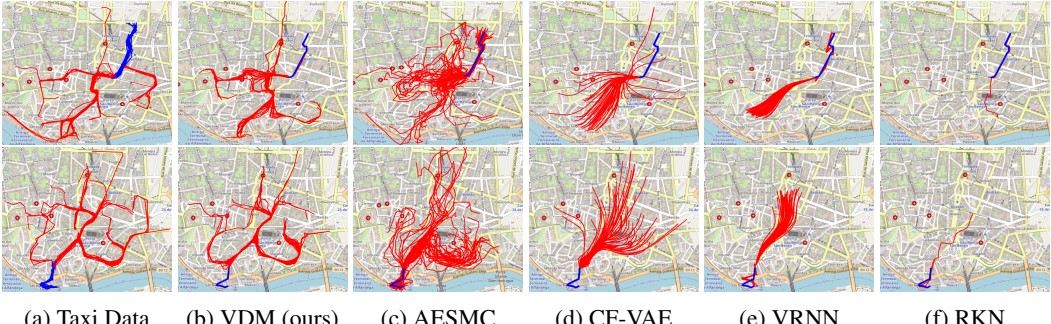

| (a) Taxi Data | (b) VDM (ours) | (c) AESMC | (d) CF-VAE | (e) VRNN | (f) RKN |

Figure 1: Forecasting taxi trajectories is challenging due to the highly multi-modal nature of the data (Fig. 1a). VDM (Fig. 1b) succeeds in generating diverse plausible predictions (red), based the beginning of a trajectory (blue). The other methods, AESMC (Le et al., 2018), CF-VAE (Bhattacharyya et al., 2019), VRNN Chung et al. (2015), RKN Becker et al. (2019), suffer from mode averaging.

approximation. We can see in Fig. 1b, that VDM trained on a dataset of taxi trajectories produces forecasts with the desired multi-modality while other methods overgeneralize.

- **An evaluation metric for multi-modal tasks.** The negative log-likelihood measures predictive accuracy but neglects an important aspect of multi-modal forecasts – sample diversity. In Section 4, we derive a score based on the Wasserstein distance (Villani, 2008) which evaluates both sample quality and diversity. This metric complements our evaluation based on log-likelihoods.

- **An extensive empirical study.** in Section 4, we use VDM to study various datasets, including a synthetic data with four modes, a stochastic Lorenz attractor, the taxi trajectories, and a U.S. pollution dataset with the measurements of various pollutants over time. We illustrate VDM's ability in modeling multi-modal dynamics, and provide quantitative comparisons to other methods showing that VDM compares favorably to previous work.

## 2 RELATED WORK

**Neural recurrent models.** Recurrent neural networks (RNNs) such as LSTMs (Hochreiter & Schmidhuber, 1997) and GRUs (Chung et al., 2014) have proven successful on many time series modeling tasks. However, as deterministic models they cannot capture uncertainties in their dynamic predictions. Stochastic RNNs make these sequence models non-deterministic (Chung et al., 2015; Fraccaro et al., 2016; Gemici et al., 2017; Li & Mandt, 2018). For example, the variational recurrent neural network (VRNN) (Chung et al., 2015) enables multiple stochastic forecasts due to its stochastic transition dynamics. An extension of VRNN (Goyal et al., 2017) uses an auxiliary cost to alleviate the KL-vanishing problem. It improves on VRNN inference by forcing the latent variables to also be predictive of future observations. Another line of related methods rely on particle filtering (Naesseth et al., 2018; Le et al., 2018; Hirt & Dellaportas, 2019) and in particular sequential Monte Carlo (SMC) to improve the evidence lower bound. In contrast, VDM adopts an explicitly multi-modal posterior approximation. Another SMC-based work (Saeedi et al., 2017) employs search-based techniques for multi-modality but is limited to models with finite discrete states. Recent works (Schmidt & Hofmann, 2018; Schmidt et al., 2019; Ziegler & Rush, 2019) use normalizing flows in the latent space to model the transition dynamics. A normalizing flow requires many layers to transform its base distribution into a truly multi-modal distribution in practice. In contrast, mixture density networks (as used by VDM) achieve multi-modality by mixing only one layer of neural networks. A task orthogonal to multi-modal inference is learning disentangled representations. Here too, mixture models are used (Chen et al., 2016; Li et al., 2017). These papers use discrete variables and a mutual information based term to disentangle different aspects of the data.

VAE-like models (Bhattacharyya et al., 2018; 2019) and GAN-like models (Sadeghian et al., 2019; Kosaraju et al., 2019) only have global, time independent latent variables. Yet, they show good results on various tasks, including forecasting. With a deterministic decoder, these models focus on average dynamics and don't capture local details (including multi-modal transitions) very well. Sequential latent variable models are described next.

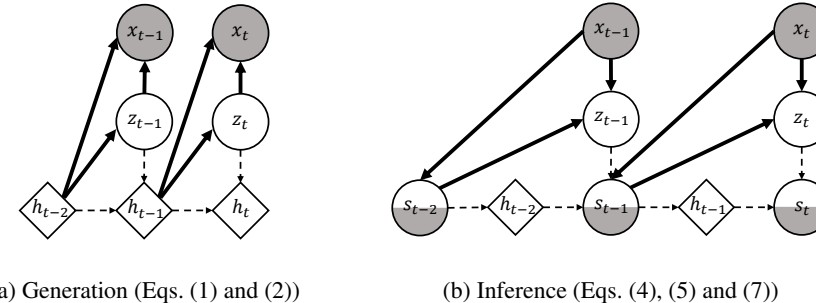

(a) Generation (Eqs. (1) and (2))          (b) Inference (Eqs. (4), (5) and (7))

Figure 2: Graphical illustrations of VDM. Dashed lines denote deterministic dependencies such as transformations, marginalization, or computing the mean, as explained in the main text, while bold lines denote stochastic dependencies. The half-shaded node for $\mathbf{s}_t$ indicates that $\mathbf{s}_t$ is being marginalized out as opposed to conditioned on.

**Deep state-space models.** Classical State-space models (SSMs) are popular due to their tractable inference and interpretable predictions. Similarly, *deep* SSMs with locally linear transition dynamics enjoy tractable inference (Karl et al., 2017; Fraccaro et al., 2017; Rangapuram et al., 2018; Becker et al., 2019). However, these models are often not expressive enough to capture complex (or highly multi-modal) dynamics. Nonlinear deep SSMs (Krishnan et al., 2017; Zheng et al., 2017; Doerr et al., 2018; De Brouwer et al., 2019; Gedon et al., 2020) are more flexible. Their inference is often no longer tractable and requires variational approximations. Unfortunately, in order for the inference model to be tractable, the variational approximations are often simplistic and don't approximate multi-modal posteriors well with negative effects on the trained models. Multi-modality can be incorporated via additional discrete switching latent variables, such as recurrent switching linear dynamical systems (Linderman et al., 2017; Nassar et al., 2018; Becker-Ehmck et al., 2019). However, these discrete states make inference more involved.

## 3 VARIATIONAL DYNAMIC MIXTURES

We develop VDM, a new sequential latent variable model for multi-modal dynamics. Given sequential observations $\mathbf{x}_{1:T} = (\mathbf{x}_1, \dots, \mathbf{x}_T)$, VDM assumes that the underlying dynamics are governed by latent states $\mathbf{z}_{1:T} = (\mathbf{z}_1, \dots, \mathbf{z}_T)$. We first present the generative process and the multi-modal inference model of VDM. We then derive a new variational objective that encourages multi-modal posterior approximations and we explain how it is regularized via hybrid-training. Finally, we introduce a new sampling method used in the inference procedure.

**Generative model.** The generative process consists of a transition model and an emission model. The transition model $p(\mathbf{z}_t \mid \mathbf{z}_{<t})$ describes the temporal evolution of the latent states and the emission model $p(\mathbf{x}_t \mid \mathbf{z}_{\leq t})$ maps the states to observations. We assume they are parameterized by two separate neural networks, the transition network $\phi^{tra}$ and the emission network $\phi^{dec}$. To give the model the capacity to capture longer range temporal correlations we parametrize the transition model with a recurrent architecture $\phi^{\text{GRU}}$ (Auger-Méthé et al., 2016; Zheng et al., 2017) such as a GRU (Chung et al., 2014). The latent states $\mathbf{z}_t$ are sampled recursively from

$$\mathbf{z}_t \mid \mathbf{z}_{<t} \sim \mathcal{N}(\mu_{0,t}, \sigma_{0,t}^2 \mathbb{I}), \quad \text{where} \quad [\mu_{0,t}, \sigma_{0,t}^2] = \phi^{tra}(\mathbf{h}_{t-1}), \quad \mathbf{h}_{t-1} = \phi^{\text{GRU}}(\mathbf{z}_{t-1}, \mathbf{h}_{t-2}), \quad (1)$$

and are then decoded such that the observations can be sampled from the emission model,

$$\mathbf{x}_t \mid \mathbf{z}_{\leq t} \sim \mathcal{N}(\mu_{x,t}, \sigma_{x,t}^2 \mathbb{I}), \quad \text{where} \quad [\mu_{x,t}, \sigma_{x,t}^2] = \phi^{dec}(\mathbf{z}_t, \mathbf{h}_{t-1}). \quad (2)$$

This generative process is similar to (Chung et al., 2015), though we did not incorporate autoregressive feedback due to its negative impact on long-term generation (Ranzato et al., 2016; Lamb et al., 2016). The competitive advantage of VDM comes from a more expressive inference model.

**Inference model.** VDM is based on a new procedure for multi-modal inference. The main idea is that to approximate the posterior at time $t$, we can use the posterior approximation of the previous

time step and exploit the generative model's transition model $\phi^{\text{GRU}}$. This leads to a sequential inference procedure. We first use the forward model to transform the approximate posterior at time $t-1$ into a distribution at time $t$. In a second step, we use samples from the resulting transformed distribution and combine each sample with data evidence $\mathbf{x}_t$, where every sample parameterizes a Gaussian mixture component. As a result, we obtain a multi-modal posterior distribution that depends on data evidence, but also on the previous time step's posterior.

In more detail, for every $\mathbf{z}_t$, we define its corresponding recurrent state as the transformed random variable $\mathbf{s}_t = \phi^{\text{GRU}}(\mathbf{z}_t, \mathbf{h}_{t-1})$, using a deterministic hidden state $\mathbf{h}_{t-1} = \mathbb{E}\left[\mathbf{s}_{t-1}\right]$. The variational family of VDM is defined as follows:

$$q(\mathbf{z}_{1:T} \mid \mathbf{x}_{1:T}) = \prod_{t=1}^{T} q(\mathbf{z}_t \mid \mathbf{x}_{\leq t}) = \prod_{t=1}^{T} \int q(\mathbf{z}_t \mid \mathbf{s}_{t-1}, \mathbf{x}_t) q(\mathbf{s}_{t-1} \mid \mathbf{x}_{\leq t}) \mathrm{d}\mathbf{s}_{t-1}. \tag{3}$$

Chung et al. (2015) also use a sequential inference procedure, but without considering the distribution of $\mathbf{s}_t$. Only a single sample is propagated through the recurrent network and all other information about the distribution of previous latent states $\mathbf{z}_{<t}$ is lost. In contrast, VDM explicitly maintains $\mathbf{s}_t$ as part of the inference model. Through marginalization, the entire distribution is taken into account for inferring the next state $\mathbf{z}_t$. Beyond the factorization assumption and the marginal consistency constraint of Eq. (3), the variational family of VDM needs two more choices to be fully specified; First, one has to choose the parametrizations of $q(\mathbf{z}_t \mid \mathbf{s}_{t-1}, \mathbf{x}_t)$ and $q(\mathbf{s}_{t-1} \mid \mathbf{x}_{\leq t})$ and second, one has to choose a sampling method to approximate the marginalization in Eq. (3). These choices determine the resulting factors $q(\mathbf{z}_t \mid \mathbf{x}_{\leq t})$ of the variational family.

We assume that the variational distribution of the recurrent state factorizes as $q(\mathbf{s}_{t-1} \mid \mathbf{x}_{\leq t}) = \omega(\mathbf{s}_{t-1}, \mathbf{x}_t)\tilde{q}(\mathbf{s}_{t-1} \mid \mathbf{x}_{<t})$, i.e. it is the distribution of the recurrent state given the past observation[2], re-weighted by a weighting function $\omega(\mathbf{s}_{t-1}, \mathbf{x}_t)$ which involves only the current observations. For VDM, we only need samples from $\tilde{q}(\mathbf{s}_{t-1} \mid \mathbf{x}_{<t})$, which are obtained by sampling from the previous posterior approximation $q(\mathbf{z}_{t-1} \mid \mathbf{x}_{<t})$ and transforming the sample with the RNN,

$$\mathbf{s}_{t-1}^{(i)} \sim \tilde{q}(\mathbf{s}_{t-1} \mid \mathbf{x}_{<t}) \quad \text{equiv. to} \quad \mathbf{s}_{t-1}^{(i)} = \phi^{\text{GRU}}(\mathbf{z}_{t-1}^{(i)}, \mathbf{h}_{t-2}), \quad \mathbf{z}_{t-1}^{(i)} \sim q(\mathbf{z}_{t-1} \mid \mathbf{x}_{<t}), \tag{4}$$

where $i$ indexes the samples. The RNN $\phi^{\text{GRU}}$ has the same parameters as in the generative model.

Augmenting the variational model with the recurrent state has another advantage; approximating the marginalization in Eq. (3) with $k$ samples from $q(\mathbf{s}_{t-1} \mid \mathbf{x}_{\leq t})$ and choosing a Gaussian parametrization for $q(\mathbf{z}_t \mid \mathbf{s}_{t-1}, \mathbf{x}_t)$ results in a q-distribution $q(\mathbf{z}_t \mid \mathbf{x}_{\leq t})$ that resembles a mixture density network (Bishop, 2006), which is a convenient choice to model multi-modal distributions.

$$q(\mathbf{z}_t \mid \mathbf{x}_{\leq t}) = \sum_{i}^{k} \omega_t^{(i)} \mathcal{N}(\mu_{z,t}^{(i)}, \sigma_{z,t}^{(i)2}\mathbb{I}), \qquad [\mu_{z,t}^{(i)}, \sigma_{z,t}^{(i)2}] = \phi^{inf}(\mathbf{s}_{t-1}^{(i)}, \mathbf{x}_t). \tag{5}$$

We assume $q(\mathbf{z}_t \mid \mathbf{s}_{t-1}, \mathbf{x}_t)$ to be Gaussian and use an inference network $\phi^{inf}$ to model the effect of the observation $\mathbf{x}_t$ and recurrent state $\mathbf{s}_{t-1}$ on the mean and variance of the mixture components.

The mixture weights $\omega_t^{(i)} := \omega(\mathbf{s}_{t-1}^{(i)}, \mathbf{x}_t)/k$ come from the variational distribution $q(\mathbf{s}_{t-1} \mid \mathbf{x}_{\leq t}) = \omega(\mathbf{s}_{t-1}, \mathbf{x}_t)\tilde{q}(\mathbf{s}_{t-1} \mid \mathbf{x}_{<t})$ and importance sampling[3]. We are free to choose how to parametrize the weights, as long as all variational distributions are properly normalized. Setting

$$\omega_t^{(i)} = \omega(\mathbf{s}_{t-1}^{(i)}, \mathbf{x}_t)/k := \mathbb{1}(i = \arg\max_{j} p(\mathbf{x}_t \mid \mathbf{h}_{t-1} = \mathbf{s}_{t-1}^{(j)})), \tag{6}$$

achieves this. In Appendix A, we explain this choice with importance sampling and in Appendix H, we compare the performance of VDM under alternative variational choices for the weights.

In the next time-step, plugging the variational distribution $q(\mathbf{z}_t \mid \mathbf{x}_{\leq t})$ into Eq. (4) yields the next distribution over recurrent states $\tilde{q}(\mathbf{s}_t \mid \mathbf{x}_{\leq t})$. For this, the expected recurrent state $\mathbf{h}_{t-1}$ is required.

---

[2]$\tilde{q}(\mathbf{s}_{t-1} \mid \mathbf{x}_{<t})$ is the distribution obtained by transforming the previous $z_{t-1} \sim q(\mathbf{z}_{t-1}|\mathbf{x}_{<t})$ through the RNN. It can be expressed analytically using the Kronecker $\delta$ to compare whether the stochastic variable $\mathbf{s}_{t-1}$ equals the output of the RNN: $\tilde{q}(\mathbf{s}_{t-1} \mid \mathbf{x}_{<t}) \propto \int \delta(\mathbf{s}_{t-1} - \phi^{\text{GRU}}(\mathbf{z}_{t-1}, \mathbf{h}_{t-2}))q(\mathbf{z}_{t-1} \mid \mathbf{x}_{t-1}, \lambda_{t-1})\mathrm{d}\mathbf{z}_{t-1}$.

[3]the $\omega$ adjusts for using samples from $\tilde{q}(\mathbf{s}_{t-1} \mid \mathbf{x}_{<t})$ when marginalizing over $\omega(\mathbf{s}_{t-1}, \mathbf{x}_t)\tilde{q}(\mathbf{s}_{t-1} \mid \mathbf{x}_{<t})$

We approximate the update using the same $k$ samples (and therefore the same weights) as in Eq. (5).

$$\mathbf{h}_{t-1} = \mathbb{E}[\mathbf{s}_{t-1}] = \int \mathbf{s}_{t-1} \, q(\mathbf{s}_{t-1} \mid \mathbf{x}_{\leq t}) \mathrm{d}\mathbf{s}_{t-1} \approx \sum_i^k \omega_t^{(i)} \mathbf{s}_{t-1}^{(i)}. \tag{7}$$

A schematic view of the generative and inference model of VDM is shown in Fig. 2. In summary, the inference model of VDM alternates between Eqs. (4) to (7). Latent states are sampled from the posterior approximation of the previous time-step and transformed by Eq. (4) into samples of the recurrent state of the RNN. These are then combined with the new observation $\mathbf{x}_t$ to produce the next variational posterior Eq. (5) and the expected recurrent state is updated (Eq. (7)). These are then used in Eq. (4) again. Approximating the marginalization in Eq. (3) with a single sample, recovers the inference model of VRNN (Chung et al., 2015), and fails in modeling multi-modal dynamics as shown in Fig. 3. In comparison, VDM's approximate marginalization over the recurrent states with multiple samples succeeds in modeling multi-modal dynamics.

**Variational objective.** We develop an objective to optimize the variational parameters of VDM $\phi = [\phi^{tra}, \phi^{dec}, \phi^{\mathrm{GRU}}, \phi^{inf}]$. The evidence lower bound (ELBO) at each time step is

$$\begin{aligned} \mathcal{L}_{\mathrm{ELBO}}(\mathbf{x}_{\leq t}, \phi) := &\frac{1}{k} \sum_i^k \omega(\mathbf{s}_{t-1}^{(i)}, \mathbf{x}_t) \mathbb{E}_{q(\mathbf{z}_t \mid \mathbf{s}_{t-1}^{(i)}, \mathbf{x}_t)} \left[ \log p(\mathbf{x}_t \mid \mathbf{z}_t, \mathbf{h}_{t-1} = \mathbf{s}_{t-1}^{(i)}) \right] \\ &+ \frac{1}{k} \sum_i^k \omega(\mathbf{s}_{t-1}^{(i)}, \mathbf{x}_t) \mathbb{E}_{q(\mathbf{z}_t \mid \mathbf{s}_{t-1}^{(i)}, \mathbf{x}_t)} \left[ \log \frac{p(\mathbf{z}_t \mid \mathbf{h}_{t-1} = \mathbf{s}_{t-1}^{(i)})}{q(\mathbf{z}_t \mid \mathbf{s}_{t-1}^{(i)}, \mathbf{x}_t)} \right] \\ &- \frac{1}{k} \sum_i^k \omega(\mathbf{s}_{t-1}^{(i)}, \mathbf{x}_t) \left[ \log \omega(\mathbf{s}_{t-1}^{(i)}, \mathbf{x}_t) + \mathbf{C} \right] \end{aligned} \tag{8}$$

**Claim 1.** *The ELBO in Eq. (8) is a lower bound on the log evidence* $\log p(\mathbf{x}_t \mid \mathbf{x}_{<t})$,

$$\log p(\mathbf{x}_t \mid \mathbf{x}_{<t}) \geq \mathcal{L}_{\mathrm{ELBO}}(\mathbf{x}_{\leq t}, \phi), \qquad \text{(see proof in Appendix B)} \,. \tag{9}$$

In addition to the ELBO, the objective of VDM has two regularization terms,

$$\mathcal{L}_{\mathrm{VDM}}(\phi) = \sum_{t=1}^{T} \mathbb{E}_{p_{\mathrm{data}}} \left[ -\mathcal{L}_{\mathrm{ELBO}}(\mathbf{x}_{\leq t}, \phi) - \omega_1 \mathcal{L}_{pred}(\mathbf{x}_{\leq t}, \phi) \right] + \omega_2 \mathcal{L}_{adv}(\mathbf{x}_{\leq t}, \phi) \,. \tag{10}$$

In an ablation study in Appendix E, we compare the effect of including and excluding the regularization terms in the objective. VDM is competitive without these terms, but we got the strongest results by setting $\omega_{1,2} = 1$ (this is the only nonzero value we tried. This hyperparameter could be tuned even further.) The first regularization term $\mathcal{L}_{pred}$, encourages the variational posterior (from the previous time step) to produce samples that maximize the predictive likelihood,

$$\mathcal{L}_{pred}(\mathbf{x}_{\leq t}, \phi) = \log \mathbb{E}_{q(\mathbf{s}_{t-1} \mid \mathbf{x}_{<t})} \left[ p(\mathbf{x}_t \mid \mathbf{s}_{t-1}, \mathbf{x}_{<t}) \right] \approx \log \frac{1}{k} \sum_i^k p(\mathbf{x}_t \mid \mathbf{s}_{t-1}^{(i)}) \,. \tag{11}$$

This regularization term is helpful to improve the prediction performance, since it depends on the predictive likelihood of samples, which isn't involved in the ELBO. The second optional regularization term $\mathcal{L}_{adv}$ (Eq. (12)) is based on ideas from hybrid adversarial-likelihood training (Grover et al., 2018; Lucas et al., 2019). These training strategies have been developed for generative models of images to generate sharper samples while avoiding "mode collapse". We adapt these ideas to generative models of dynamics. The adversarial term $\mathcal{L}_{adv}$ uses a forward KL-divergence, which enables "quality-driven training" to discourage probability mass in spurious areas.

$$\mathcal{L}_{adv}(\mathbf{x}_{\leq t}, \phi) = \mathcal{D}_{\mathrm{KL}}(p(\mathbf{x}_t \mid \mathbf{x}_{<t}) \| p_{\mathcal{D}}(\mathbf{x}_t \mid \mathbf{x}_{<t})) = \mathbb{E} \left[ \log p(\mathbf{x}_t \mid \mathbf{x}_{<t}) - \log p_{\mathcal{D}}(\mathbf{x}_t \mid \mathbf{x}_{<t}) \right] \tag{12}$$

The expectation is taken w.r.t. $p(\mathbf{x}_t \mid \mathbf{x}_{<t})$. The true predictive distribution $p_{\mathcal{D}}(\mathbf{x}_t \mid \mathbf{x}_{<t})$ is unknown. Instead, we can train the generator of a conditional GAN (Mirza & Osindero, 2014), while assuming an optimal discriminator. As a result, we optimize Eq. (12) in an adversarial manner, conditioning on $\mathbf{x}_{<t}$ at each time step. Details about the discriminator are in Appendix G.

**Stochastic cubature approximation (SCA).** The variational family of VDM is defined by a number of modeling choices, including the factorization and marginal consistency assumptions of Eq. (3), the parametrization of the transition and inference networks Eqs. (4) and (5), and the choice of weighting function $\omega(\cdot)$. It is also sensitive to the choice of sampling method which we discuss here. In principle, we could use Monte-Carlo methods. However, for a relatively small number of samples $k$, Monte-Carlo methods don't have a mechanism to control the quality of samples. We instead develop a semi-stochastic approach based on the cubature approximation (Wan & Van Der Merwe, 2000; Wu et al., 2006; Arasaratnam & Haykin, 2009), which chooses samples more carefully. The cubature approximation proceeds by constructing $k = 2d + 1$ so-called sigma points, which are optimally spread out on the $d$-dimensional Gaussian with the same mean and covariance as the distribution we need samples from. In SCA, the deterministic sigma points are infused with Gaussian noise to obtain stochastic *sigma variables*. A detailed derivation of SCA is in Appendix D.

We use SCA for various reasons: First, it typically requires fewer samples than Monte-Carlo methods because the sigma points are carefully chosen to capture the first two moments of the underlying distribution. Second, it ensures a persistence of the mixture components; when we resample, we sample another nearby point from the mixture component and not an entirely new location.

## 4 EVALUATION AND EXPERIMENTS

In this empirical study, we evaluate VDM's ability to model multi-modal dynamics and show its competitive forecasting performance in various domains. We first introduce the evaluation metrics and baselines. Experiments on synthetic data demonstrate that VDM is truly multi-modal thereby supporting the modeling choices of Section 3, especially for the inference model. Then, experiments on real-world datasets with challenging multi-modal dynamics show the benefit of VDM over state-of-the art (deep) probabilistic time-series models.

**Evaluation metrics.** In the experiments, we always create a training set, a validation set, and a test set. During validation and test, each trajectory is split into two parts; initial observations (given to the models for inference) and continuations of the trajectories (to be predicted and not accessible to the models). The inference models are used to process the initial observations and to infer latent states. These are then processed by the generative models to produce forecasts.

We use 3 criteria to evaluate these forecasts (i) multi-steps ahead prediction $p(\mathbf{x}_{t+1:t+\tau} \mid \mathbf{x}_{1:t})$, (ii) one-step-ahead prediction $p(\mathbf{x}_{t+1} \mid \mathbf{x}_{1:t})$, and (iii) empirical Wasserstein distance. As in other work (Lee et al., 2017; Bhattacharyya et al., 2018; 2019), (i) and (ii) are reported in terms of negative log-likelihood. While the predictive distribution for one-step-ahead prediction is in closed-form, the long-term forecasts have to be computed using samples. For each ground truth trajectory $\mathbf{x}$ we generate $n = 1000$ forecasts $\hat{\mathbf{x}}_i$ given initial observations from the beginning of the trajectory

$$NLL = -\log\left(\frac{1}{n}\sum_i^n \frac{1}{\sqrt{2\pi}}\exp\left(-\frac{(\hat{\mathbf{x}}_i - \mathbf{x})^2}{2}\right)\right), \tag{13}$$

This evaluates the predictive accuracy but neglects a key aspect of multi-modal forecasts – diversity.

We propose a new evaluation metric, which takes both diversity and accuracy of predictions into account. It relies on computing the Wasserstein distance between two empirical distributions $P, Q$

$$W(P, Q) = \inf_\pi\left(\frac{1}{n}\sum_i^n \|(\mathbf{x}_i - \mathbf{y}_{\pi(i)}\|_2\right), \tag{14}$$

where $\mathbf{x}$ and $\mathbf{y}$ are the discrete samples of $P$ and $Q$, and $\pi$ denotes all permutations (Villani, 2008). To use this as an evaluation measure for multi-modal forecasts, we do the following. We select $n$ samples from the test set with similar initial observations. If the dynamics in the data are multi-modal the continuations of those $n$ trajectories will be diverse and this should be reflected in the forecasts. For each of the $n$ samples, the model generates 10 forecasts and we get $n$ groups of samples. With Eq. (14) the empirical W-distance between the $n$ true samples, and each group of generated samples can be calculated. The averaged empirical W-distance over groups evaluates how well the generated samples match the ground truth. Repeating this procedure with different initial trajectories evaluates the distance between the modeled distribution and the data distribution.

$$\text{VDM}(k = 9) \qquad\qquad \text{VDM}(k = 1) \qquad\qquad \text{AESMC}(k = 9)$$

$$\tilde{\mathbf{D}} \qquad p(\mathbf{z}_2|\mathbf{D}) \quad p(\mathbf{z}_2|\mathbf{x}_{\leq 1}) \qquad\qquad \tilde{\mathbf{D}} \qquad p(\mathbf{z}_2|\mathbf{D}) \quad p(\mathbf{z}_2|\mathbf{x}_{\leq 1}) \qquad\qquad \tilde{\mathbf{D}} \qquad p(\mathbf{z}_2|\mathbf{D}) \quad p(\mathbf{z}_2|\mathbf{x}_{\leq 1})$$

Figure 3: **Experiments on 2d synthetic data with 4 modes highlight the multi-modality of VDM.**
We train VDM($k = 9$) (left), VDM($k = 1$) (middle), and AESMC($k = 9$) (right) on a training set
of trajectories $\mathbf{D}$ of length 4, and plot generated trajectories $\tilde{\mathbf{D}}$ (2 colors for 2 dimensions). We also
plot the aggregated posterior $p(\mathbf{z}_2|\mathbf{D})$, and the predictive prior $p(\mathbf{z}_2|\mathbf{x}_{\leq 1})$ (4 colors for 4 clusters,
and not related to the colors in the trajectories plot) at the second time step. Only VDM learns a
multi-modal predictive prior, which explains its success in modeling multi-modal dynamics.

Table 1: Prediction error on stochastic Lorenz attractor for three evaluation metrics (details in main
text). VDM($k = 13$) achieves the best performance, and AESMC also gives comparable results.

| | RKN | VRNN | CF-VAE | AESMC | VDM($k = 1$) | VDM($k = 13$) |
|---|---|---|---|---|---|---|
| Multi-steps | 104.41 | 65.89±0.21 | 32.41±0.13 | 25.01±0.22 | 25.03±0.28 | **24.46±0.12** |
| One-step | 1.88 | -1.63 | n.a | -1.69 | -1.81 | **-1.81** |
| W-distance | 16.16 | 16.14±0.006 | 8.44±0.005 | 7.29±0.005 | 7.31±0.002 | **7.28±0.002** |

**Baselines.** We choose baselines from three classes of models. Two stochastic recurrent models
are variational recurrent neural network (VRNN) (Chung et al., 2015) and auto-encoding sequential
Monte Carlo (AESMC) (Le et al., 2018). VRNN has a similar but more powerful generative model
than VDM, and AESMC uses SMC to achieve a tighter lower bound. But compared to VDM, both
methods have a less powerful inference model which limits their capacity to capture multi-modal
distributions. The third baseline is a deep SSM. The recurrent Kalman network (RKN) (Becker
et al., 2019) models the latent space with a locally linear SSMs, which makes the prediction step and
update step analytic (as for Kalman filters (Kalman, 1960)). A final baseline is the conditional flow
variational autoencoder (CF-VAE) (Bhattacharyya et al., 2019), which uses conditional normalizing
flows to model a global prior for the future continuations and achieves state-of-the-art performances.

To investigate the necessity of taking multiple samples in the VDM inference model, we also com-
pared to VDM($k = 1$) which uses only a single sample in Eq. (5). VDM($k = 1$) has a simpler
generative model than VRNN (it considers no autoregressive feedback of the observations $\mathbf{x}$), but
the same inference model. More ablations for the modeling choices of VDM are in Appendix H.

For fair comparison, we fix the dimension of the latent variables $\mathbf{z}_t$ and $\mathbf{h}_t$ to be the same for VDM,
AESMC, and VRNN which have the same resulting model size (except for the additional autore-
gressive feedback in VRNN). AESMC and VDM always use the same number of particles/samples.
RKN does not have recurrent states, so we choose a higher latent dimension to make model size
comparable. In contrast, CF-VAE has only one global latent variable which needs more capacity
and we make it higher-dimensional than $\mathbf{z}_t$. Details for each experiment are in Appendix G.

**Synthetic data with multi-modal dynamics.** We generate synthetic data with two dimensions and
four modes and compare the performance of VDM with 9 samples (Fig. 3, left), VDM with a single
sample (Fig. 3, middle), and AESMC using 9 particles (Fig. 3, right). Since variational inference
is known to try to match the aggregated posterior with the predictive prior (Tomczak & Welling,
2018), it is instructive to fit all three models and to look at their predictive prior $p(\mathbf{z}_2|\mathbf{x}_{\leq 1})$ and the
aggregated posterior $p(\mathbf{z}_2|\mathbf{D})$. Because of the multi-modal nature of the problem, all 3 aggregated
posteriors are multi-modal, but only VDM($k = 9$) learns a multi-modal predictive prior (thanks to
its choice of variational family). Although AESMC achieves a good match between the prior and the
aggregated posterior, the predictive prior does not clearly separate into different modes. In contrast,
the inference model of VDM successfully uses the weights (Eq. (6)), which contain information
about the incoming observation, to separate the latent states into separate modes.

**Stochastic Lorenz attractor.** The Lorenz attractor is a system governed by ordinary differential
equations. We add noise to the transition and emission function to make it stochastic (details in
Appendix F.1). Under certain parameter settings it is chaotic – even small errors can cause consid-
erable differences in the future. This makes forecasting its dynamics very challenging. All models
are trained and then tasked to predict 90 future observations given 10 initial observations. Fig. 4

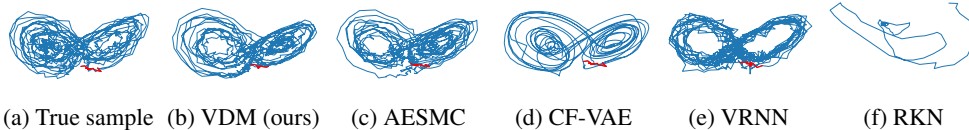

(a) True sample (b) VDM (ours) (c) AESMC (d) CF-VAE (e) VRNN (f) RKN

Figure 4: Generated samples from VDM and baselines for stochastic Lorenz attractor. The models generate the remaining 990 observations (blue) based on the first 10 observations (red). Due to the chaotic property, the reconstruction is impossible even the model learns the right dynamics. VDM and AESMC capture the dynamics very well, while RKN fails in capturing the stochastic dynamics.

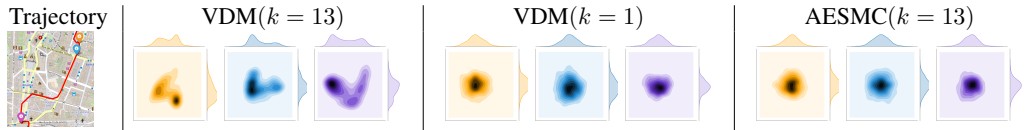

Figure 5: An illustration of predictive priors $p(\mathbf{z}_t|\mathbf{x}_{<t})$ of taxi trajectories from VDM($k = 13$), VDM($k = 1$), and AESMC($k = 13$) at 3 forks in the road marked on the map (left). VDM($k = 13$) succeeds in capturing the multi-modal distributions, while the other methods approximate them with uni-modal distributions. For visualization, the distributions have been projected to 2d with KDE.

Table 2: Prediction error on taxi trajectories for three evaluation metrics (details in main text). CF-VAE gives the best result in multi-steps prediction, since it uses one global latent variable, while sequential models rely on a sequence of local latent variables. Meanwhile, VDM($k = 13$) outperforms all sequential models, and performs better in other metrics than CF-VAE.

|  | RKN | VRNN | CF-VAE | AESMC | VDM($k=1$) | VDM($k=13$) |
|---|---|---|---|---|---|---|
| Multi-steps | 4.25 | 5.51±0.002 | **2.77**±0.001 | 3.31±0.001 | 3.26±0.001 | 2.85±0.002 |
| One-step | -2.90 | -2.77 | n.a | -2.87 | -2.99 | **-3.62** |
| W-distance | 2.07 | 2.43±0.0002 | 0.76±0.0003 | 0.66±0.0004 | 0.69±0.0005 | **0.56**±0.0005 |

illustrates qualitatively that VDM (Fig. 4b) and AESMC (Fig. 4c) succeed in modeling the chaotic dynamics of the stochastic Lorenz attractor, while CF-VAE (Fig. 4d) and VRNN (Fig. 4e) miss local details, and RKN (Fig. 4f) which lacks the capacity for stochastic transitions does not work at all. VDM achieves the best scores on all metrics (Table 1). Since the dynamics of the Lorenz attractor are governed by ordinary differential equations, the transition dynamics at each time step are not obviously multi-modal, which explains why all models with stochastic transitions do reasonably well. Next, we will show the advantages of VDM on real-world data with multi-modal dynamics.

**Taxi trajectories.** The taxi trajectory dataset involves taxi trajectories with variable lengths in Porto, Portugal. Each trajectory is a sequence of two dimensional locations over time. Here, we cut the trajectories to a fixed length of 30 to simplify the comparison (details in Appendix F.2). The task is to predict the next 20 observations given 10 initial observations. Ideally, the forecasts should follow the street map (though the map is not accessible to the models).

The results in Table 2 show that VDM outperforms the other *sequential* latent variable models in all evaluations. However, it turns out that for multi-step forecasting learning global structure is advantageous, and CF-VAE which is a global latent variable model, achieves the highest results. However, this value doesn't match the qualitative results in Fig. 1. Since CF-VAE has to encode the entire structure of the trajectory forcast into a single latent variable, its predictions seem to average over plausible continuations but are locally neither plausible nor accurate. In comparison, VDM and the other models involve a sequence of latent variables. As the forecasting progresses, the methods update their distribution over latest states, and the impact of the initial observations becomes weaker and weaker. As a result, local structure is captured more accurately. While the forecasts are plausible and can be highly diverse, they potentially evolve into other directions than the ground truth. For this reason, their multi-step prediction results are worse in terms of log-likelihood. That's why the empirical W-distance is useful to complement the evaluation of multi-modal tasks. It reflects that the forecasts of VDM are diverse and plausible. Additionally, we illustrate the predictive prior $p(\mathbf{z}_t|\mathbf{x}_{<t})$ at different time steps in Fig. 5. VDM($k = 13$) learns a multi-modal predictive prior, which VDM($k = 1$) and AESMC approximate it with an uni-modal Gaussian.

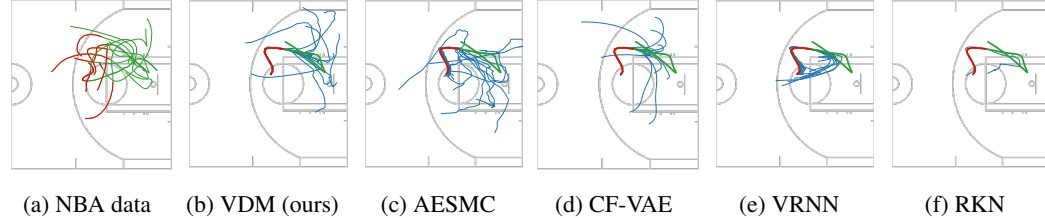

(a) NBA data    (b) VDM (ours)    (c) AESMC    (d) CF-VAE    (e) VRNN    (f) RKN

Figure 6: VDM and CF-VAE generate plausible multi-modal trajectories of basketball plays. Each model's forecasts (blue) are based on the first 10 observations (red). Ground truth data is green.

Table 3: Prediction error on U.S. pollution data for two evaluation metrics (details in main text). VDM makes the most accurate multi-step and one-step predictions.

|  | RKN | VRNN | CF-VAE | AESMC | VDM($k = 1$) | VDM($k = 17$) |
|---|---|---|---|---|---|---|
| Multi-steps | 53.13 | 49.32±0.13 | 45.86±0.04 | 41.14±0.13 | 42.33±0.11 | **36.72**±0.08 |
| One-step | 6.98 | 8.69 | n.a | 6.93 | 7.97 | **6.05** |

Table 4: Prediction error on basketball players' trajectories (details in main text). VDM makes the most accurate multi-step and one-step predictions.

|  | RKN | VRNN | CF-VAE | AESMC | VDM($k = 1$) | VDM($k = 13$) |
|---|---|---|---|---|---|---|
| Multi-steps | 4.88 | 5.42±0.009 | 3.24±0.003 | 3.74±0.003 | 3.56±0.005 | **3.18**±0.005 |
| One-step | 1.55 | -2.78 | n.a | -3.91 | -4.26 | **-5.18** |

**U.S. pollution data.** In this experiment, we study VDM on the U.S. pollution dataset (details in Appendix F.3). The data is collected from counties in different states from 2000 to 2016. Each observation has 12 dimensions (mean, max value, and air quality index of NO2, O3, SO2, and O3). The goal is to predict monthly pollution values for the coming 18 months, given observations of the previous six months. We ignore the geographical location and time information to treat the development tendency of pollution in different counties and different times as i.i.d.. The unknown context information makes the dynamics multi-modal and challenging to predict accurately. Due to the small size and high dimensionality of the dataset, there are not enough samples with very similar initial observations. Thus, we cannot evaluate empirical W-distance in this experiment. In multi-step predictions and one-step predictions, VDM outperforms the other methods.

**NBA SportVu data.** This dataset[4] of sequences of 2D coordinates describes the movements of basketball players and the ball. We extract the trajectories and cut them to a fixed length of 30 to simplify the comparisons (details in Appendix F.4). The task is to predict the next 20 observations given 10 initial observations. Players can move anywhere on the court and hence their movement is less structured than the taxi trajectories which are constrained by the underlying street map. Due to this, the initial movement patters are not similar enough to each other to evaluate empirical W-distance. In multi-step and one-step predictions, VDM outperforms the other baselines (Table 4). Fig. 6 illustrates qualitatively that VDM (Fig. 6b) and CF-VAE (Fig. 6d) succeed in capturing the multi-modal dynamics. The forecasts of AESMC (Fig. 6c) are less plausible (not as smooth as data), and VRNN (Fig. 6e) and RKN (Fig. 6f) fail in capturing the multi-modality.

## 5 CONCLUSION

We have presented variational dynamic mixtures (VDM), a sequential latent variable model for multi-modal dynamics. The main contribution is a new variational family. It propagates multiple samples through an RNN to parametrize the posterior approximation with a mixture density network. Additionally, we have introduced the empirical Wasserstein distance for the evaluation of multi-modal forecasting tasks, since it accounts for forecast accuracy and diversity. VDM succeeds in learning challenging multi-modal dynamics and outperforms existing work in various applications.

---

[4]A version of the dataset is available at https://www.stats.com/data-science/

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

## A   SUPPLEMENTARY TO WEIGHTING FUNCTION

In this Appendix we give intuition for our choice of weighting function Eq. (6). Since we approximate the integrals in Eqs. (3) and (7) with samples from $\tilde{q}(\mathbf{s}_{t-1} \mid \mathbf{x}_{<t})$ [5] instead of samples from $q(\mathbf{s}_{t-1} \mid \mathbf{x}_{\leq t})$, importance sampling tells us that the weigths should be

$$\omega(\mathbf{s}_{t-1}, \mathbf{x}_t) = \frac{q(\mathbf{s}_{t-1} \mid \mathbf{x}_{\leq t})}{\tilde{q}(\mathbf{s}_{t-1} \mid \mathbf{x}_{<t})} = \frac{q(\mathbf{x}_t \mid \mathbf{s}_{t-1}, \mathbf{x}_{<t})}{q(\mathbf{x}_t \mid \mathbf{x}_{<t})} \frac{\tilde{q}(\mathbf{s}_{t-1} \mid \mathbf{x}_{<t})}{\tilde{q}(\mathbf{s}_{t-1} \mid \mathbf{x}_{<t})}$$

$$= \frac{q(\mathbf{x}_t \mid \mathbf{s}_{t-1}, \mathbf{x}_{<t})}{q(\mathbf{x}_t \mid \mathbf{x}_{<t})} \propto q(\mathbf{x}_t \mid \mathbf{s}_{t-1}, \mathbf{x}_{<t}) \quad (15)$$

This is consistent with out earlier definition of $q(\mathbf{s}_{t-1} \mid \mathbf{x}_{\leq t}) = \omega(\mathbf{s}_{t-1}, \mathbf{x}_t)\tilde{q}(\mathbf{s}_{t-1} \mid \mathbf{x}_{<t})$. The weights are proportional to the likelihood of the variational model $q(\mathbf{x}_t \mid \mathbf{s}_{t-1}, \mathbf{x}_{<t})$. We choose to parametrize it using the likelihood of the generative model $p(\mathbf{x}_t \mid \mathbf{h}_{t-1} = \mathbf{s}_{t-1})$ and get

$$\omega_t^{(i)} = \omega(\mathbf{s}_{t-1}^{(i)}, \mathbf{x}_t)/k := \mathbb{1}(i = \arg\max_j p(\mathbf{x}_t \mid \mathbf{h}_{t-1} = \mathbf{s}_{t-1}^{(j)})). \quad (16)$$

With this choice of the weighting function, only the mixture component with the highest likelihood is selected to be in charge of modeling the current observation $\mathbf{x}_t$. As a result, other mixture components have the capacity to focus on different modes. This helps avoid the effect of mode-averaging. An alternative weight function is given in Appendix H.

## B   SUPPLEMENTARY TO LOWER BOUND

**Claim.** *The ELBO in Eq. (8) is a lower bound on the log evidence* $\log p(\mathbf{x}_t \mid \mathbf{x}_{<t})$,

$$\log p(\mathbf{x}_t \mid \mathbf{x}_{<t}) \geq \mathcal{L}_{\text{ELBO}}(\mathbf{x}_{\leq t}, \phi). \quad (17)$$

*Proof.* We write the data evidence as the double integral over the latent variables $\mathbf{z}_t$, and $\mathbf{z}_{<t}$.

$$\log p(\mathbf{x}_t \mid \mathbf{x}_{<t}) = \log \iint p(\mathbf{x}_t \mid \mathbf{z}_{\leq t}, \mathbf{x}_{<t})p(\mathbf{z}_t \mid \mathbf{z}_{<t}, \mathbf{x}_{<t})p(\mathbf{z}_{<t} \mid \mathbf{x}_{<t})\mathrm{d}\mathbf{z}_t\mathrm{d}\mathbf{z}_{<t} \quad (18)$$

We multiply the posterior at the previous time step $p(\mathbf{z}_{<t} \mid \mathbf{x}_{<t})$ with the ratio of the approximated posterior $\frac{q(\mathbf{z}_{<t}|\mathbf{x}_{<t})}{q(\mathbf{z}_{<t}|\mathbf{x}_{<t})}$ and the ratio $\frac{f(\mathbf{a},\mathbf{b})}{f(\mathbf{a},\mathbf{b})}$, where $f$ is any suitable function of two variables $\mathbf{a}$ and $\mathbf{b}$. The following equality holds, since the ratios equal to one.

$$\log p(\mathbf{x}_t \mid \mathbf{x}_{<t})$$
$$= \log \int \frac{f(\mathbf{a}, \mathbf{b})}{f(\mathbf{a}, \mathbf{b})} \frac{q(\mathbf{z}_{<t} \mid \mathbf{x}_{<t})}{q(\mathbf{z}_{<t} \mid \mathbf{x}_{<t})} p(\mathbf{z}_{<t} \mid \mathbf{x}_{<t}) \int p(\mathbf{x}_t \mid \mathbf{z}_{\leq t}, \mathbf{x}_{<t})p(\mathbf{z}_t \mid \mathbf{z}_{<t}, \mathbf{x}_{<t})\mathrm{d}\mathbf{z}_t\mathrm{d}\mathbf{z}_{<t} \quad (19)$$

We move the integral over $\mathbf{z}_{<t}$ with respect to $f(\mathbf{a}, \mathbf{b})q(\mathbf{z}_{<t} \mid \mathbf{x}_{<t})$ out of the log operation with applying the Jensen's inequality.

$$\log p(\mathbf{x}_t \mid \mathbf{x}_{<t}) \geq \mathbb{E}_{f(\mathbf{a},\mathbf{b})q(\mathbf{z}_{<t}|\mathbf{x}_{<t})}\left[\log \int p(\mathbf{x}_t \mid \mathbf{z}_{\leq t}, \mathbf{x}_{<t})p(\mathbf{z}_t \mid \mathbf{z}_{<t}, \mathbf{x}_{<t})\mathrm{d}\mathbf{z}_t\right] \quad (20)$$
$$- \mathbb{E}_{f(\mathbf{a},\mathbf{b})q(\mathbf{z}_{<t}|\mathbf{x}_{<t})}\left[\log f(\mathbf{a}, \mathbf{b}) + \log \frac{q(\mathbf{z}_{<t} \mid \mathbf{x}_{<t})}{p(\mathbf{z}_{<t} \mid \mathbf{x}_{<t})}\right]$$

We introduce the variational posterior $q(\mathbf{z}_t \mid \mathbf{z}_{<t}, \mathbf{x}_{\leq t})$, and apply Jensen's inequality to replace the intractable integral $\log \int p(\mathbf{x}_t \mid \mathbf{z}_{\leq t}, \mathbf{x}_{<t})p(\mathbf{z}_t \mid \mathbf{z}_{<t}, \mathbf{x}_{<t})\mathrm{d}\mathbf{z}_t$ with its lower bound.

$$\log p(\mathbf{x}_t \mid \mathbf{x}_{<t}) \geq \mathbb{E}_{f(\mathbf{a},\mathbf{b})q(\mathbf{z}_{<t}|\mathbf{x}_{<t})}\left[\mathbb{E}_{q(\mathbf{z}_t|\mathbf{z}_{<t},\mathbf{x}_{\leq t})}\left[\log \frac{p(\mathbf{x}_t \mid \mathbf{z}_{\leq t}, \mathbf{x}_{<t})p(\mathbf{z}_t \mid \mathbf{z}_{<t}, \mathbf{x}_{<t})}{q(\mathbf{z}_t \mid \mathbf{z}_{<t}, \mathbf{x}_{\leq t})}\right]\right]$$
$$- \mathbb{E}_{f(\mathbf{a},\mathbf{b})q(\mathbf{z}_{<t}|\mathbf{x}_{<t})}\left[\log f(\mathbf{a}, \mathbf{b}) + \log \frac{q(\mathbf{z}_{<t} \mid \mathbf{x}_{<t})}{p(\mathbf{z}_{<t} \mid \mathbf{x}_{<t})}\right]. \quad (21)$$

---

[5]The $\sim$ just helps to visually distinguish the two distributions that appear in the main text.

The expectation with respect to $f(\mathbf{a}, \mathbf{b})q(\mathbf{z}_{<t} \mid \mathbf{x}_{<t})$ is approximated with samples. Instead of re-sampling the entire history, samples from previous time steps are reused (they have been aggregated by the RNN) and we sample according to Eq. (4). We plugg in the weighting function $\omega(\mathbf{s}_{t-1}^{(i)}, \mathbf{x}_t)$ for $f(\mathbf{a}, \mathbf{b})$. The term $\log \frac{q(\mathbf{z}_{<t}|\mathbf{x}_{<t})}{p(\mathbf{z}_{<t}|\mathbf{x}_{<t})}$ is not affected by the incoming observation $\mathbf{x}_t$ and can be treated as a constant.

In this step, we plug in our generative model and inference model as they are described in the main text for $p$ and $q$. The conditional independence assumptions can be read of Fig. 2. In the generative model $\mathbf{h}_{t-1}$ and in the inference model $\mathbf{s}_{t-1}$ summarize the dependencies of $\mathbf{z}_t$ on the previous latent variables $\mathbf{z}_{<t}$ and observations $\mathbf{x}_{<t}$. In other words, we assume $\mathbf{z}_t$ is conditionally independent on $\mathbf{z}_{<t}$ and $\mathbf{x}_{<t}$ given $\mathbf{s}_{t-1}^{(i)}$ in the inference model (or given $\mathbf{h}_{t-1}$ in the generative model).

$$
\begin{aligned}
\log p(\mathbf{x}_t \mid \mathbf{x}_{<t}) \geq & \frac{1}{k} \sum_i^k \omega(\mathbf{s}_{t-1}^{(i)}, \mathbf{x}_t) \mathbb{E}_{q(\mathbf{z}_t|\mathbf{s}_{t-1}^{(i)}, \mathbf{x}_t)} \left[ \log p(\mathbf{x}_t \mid \mathbf{z}_t, \mathbf{h}_{t-1} = \mathbf{s}_{t-1}^{(i)}) \right] \\
& + \frac{1}{k} \sum_i^k \omega(\mathbf{s}_{t-1}^{(i)}, \mathbf{x}_t) \mathbb{E}_{q(\mathbf{z}_t|\mathbf{s}_{t-1}^{(i)}, \mathbf{x}_t)} \left[ \log \frac{p(\mathbf{z}_t \mid \mathbf{h}_{t-1} = \mathbf{s}_{t-1}^{(i)})}{q(\mathbf{z}_t \mid \mathbf{s}_{t-1}^{(i)}, \mathbf{x}_t)} \right] \\
& - \frac{1}{k} \sum_i^k \omega(\mathbf{s}_{t-1}^{(i)}, \mathbf{x}_t) \left[ \log \omega(\mathbf{s}_{t-1}^{(i)}, \mathbf{x}_t) + \mathbf{C} \right]
\end{aligned}
\tag{22}
$$

$\square$

## C  ALGORITHMS OF GENERATIVE MODEL AND INFERENCE MODEL

---
**Algorithm 1** Generative model

---
**Inputs:** $[\mu_{z,\tau}, \sigma_{z,\tau}^2], \mathbf{h}_{\tau-1}$
**Outputs:** $\mathbf{x}_{\tau+1:T}$
$\mathbf{z}_\tau \sim \mathcal{N}(\mu_{z,\tau}, \sigma_{z,\tau}^2 \mathbb{I})$
$\mathbf{h}_\tau = \phi^{\text{GRU}}(\mathbf{z}_\tau, \mathbf{h}_{\tau-1})$
**for** $t = \tau + 1 : T$ **do**
  $[\mu_{0,t}, \sigma_{0,t}^2] = \phi^{tra}(\mathbf{h}_{t-1})$
  $\mathbf{z}_t \sim \mathcal{N}(\mu_{0,t}, \sigma_{0,t}^2 \mathbb{I})$
  $\mathbf{h}_t = \phi^{\text{GRU}}(\mathbf{z}_t, \mathbf{h}_{t-1})$
  $[\mu_{x,t}, \sigma_{x,t}^2] = \phi^{dec}(\mathbf{z}_t, \mathbf{h}_{t-1})$
  $\mathbf{x}_t \sim \mathcal{N}(\mu_{x,t}, \sigma_{x,t}^2 \mathbb{I})$
**end for**

---

---
**Algorithm 2** Inference model

---
**Inputs:** $\mathbf{x}_{1:\tau}, \mathbf{h}_0$
**Outputs:** $[\mu_{z,1:\tau}, \sigma_{z,1:\tau}^2], \mathbf{h}_{\tau-1}$
$[\mu_{z,1}, \sigma_{z,1}^2] = \phi^{inf}(\mathbf{h}_0, \mathbf{x}_1)$
**for** $t = 2 : \tau$ **do**
  $\mathbf{z}_{t-1}^{(i)} \sim \mathcal{N}(\mu_{z,t-1}, \sigma_{z,t-1}^2 \mathbb{I})$
  $\mathbf{s}_{t-1}^{(i)} = \phi^{\text{GRU}}(\mathbf{z}_{t-1}^{(i)}, \mathbf{h}_{t-2})$
  $[\mu_{z,t}^{(i)}, \sigma_{z,t}^{(i)2}] = \phi^{inf}(\mathbf{s}_{t-1}^{(i)}, \mathbf{x}_t)$
  $\omega_t^{(i)} := \mathbb{1}(i = \arg\max_j p(\mathbf{x}_t \mid \mathbf{h}_{t-1} = \mathbf{s}_{t-1}^{(j)}))$
  $[\mu_{z,t}, \sigma_{z,t}^2] = \sum_i^k \omega_t^{(i)} \mathcal{N}(\mu_{z,t}^{(i)}, \sigma_{z,t}^{(i)2} \mathbb{I})$
  $\mathbf{h}_{t-1} \approx \sum_i^k \omega_t^{(i)} \mathbf{s}_{t-1}^{(i)}$
**end for**

---

## D  SUPPLEMENTARY TO STOCHASTIC CUBATURE APPROXIMATION

**Cubature approximation.**  The cubature approximation is widely used in the engineering community as a deterministic method to numerically integrate a nonlinear function $f(\cdot)$ of Gaussian random variable $z \sim \mathcal{N}(\mu_{\mathbf{z}}, \sigma_{\mathbf{z}}^2 \mathbb{I})$, with $\mathbf{z} \in \mathbb{R}^d$. The method proceeds by constructing $2d+1$ sigma points $\mathbf{z}^{(i)} = \mu_{\mathbf{z}} + \sigma_{\mathbf{z}} \xi^{(i)}$. The cubature approximation is simply a weighted sum of the sigma points propagated through the nonlinear function $f(\cdot)$,

$$
\int f(\mathbf{z}) \mathcal{N}(\mathbf{z} \mid \mu_{\mathbf{z}}, \sigma_{\mathbf{z}}^2 \mathbb{I}) \mathrm{d}\mathbf{z} \approx \sum_{i=1}^{2d+1} \gamma^{(i)} f(\mathbf{z}^{(i)}).
\tag{23}
$$

Simple analytic formulas determine the computation of weights $\gamma^{(i)}$ and the locations of $\xi^{(i)}$.

$$\gamma^{(i)} = \begin{cases} \frac{1}{2(n+\kappa)} & , i = 1, ..., 2n \\ \frac{\kappa}{n+\kappa} & , i = 0 \end{cases} \qquad \xi^{(i)} = \begin{cases} \sqrt{n+\kappa}\mathbf{e}_i & , i = 1, ..., n \\ -\sqrt{n+\kappa}\mathbf{e}_{i-n} & , i = n+1, ..., 2n \\ 0 & , i = 0 , \end{cases} \tag{24}$$

where $\kappa$ is a hyperparameter controlling the spread of the sigma points in the $n$-dimensional sphere. Further $\mathbf{e}_i$ represents a basis in the $n$-dimensional space, which is choosen to be a unit vector in cartesian space, e.g. $\mathbf{e}_1 = [1, 0, ..., 0]$.

**Stochastic cubature approximation.** In SCA, we adopt the computation of $\xi^{(i)}$ in Eq. (24), and infuse the sigma points with standard Gaussian noise $\epsilon \sim \mathcal{N}(0, \mathbb{I})$ to obtain stochastic *sigma variables* $\mathbf{s}^{(i)} = \mu_{\mathbf{z}} + \sigma_{\mathbf{z}}(\xi^{(i)} + \epsilon)$. We choose $\kappa = 0.5$ to set the weights $\gamma^{(i)}$ equally.

## E    SUPPLEMENTARY TO ABLATION STUDY OF REGULARIZATION TERMS

We investigate the effect of the regularization terms using the synthetic data from Fig. 3. We can see in Table 5, VDM($k = 9$) can be trained successfully with $\mathcal{L}_{\text{ELBO}}$ only, and both regularization terms improve the performance (negative log-likelihood of multi-steps ahead prediction), while VDM($k = 1$) doesn't work whatever the regularization terms. Additionally, we tried to train the model only with the regularization terms (each separate or together) but these options diverged during training.

Table 5: Ablation study of the regularization terms for synthetic data from Fig. 3

|  | $\mathcal{L}_{\text{ELBO}}$ | $\mathcal{L}_{\text{ELBO}}\&\mathcal{L}_{pred}$ | $\mathcal{L}_{\text{ELBO}}\&\mathcal{L}_{adv}$ | $\mathcal{L}_{\text{VDM}}$ |
|---|---|---|---|---|
| **VDM($k = 9$)** | 2.439±0.005 | 2.379±0.008 | 2.381±0.006 | **2.363**±0.004 |
| **VDM($k = 1$)** | 3.756±0.003 | 3.960±0.008 | 3.743±0.005 | 3.878±0.007 |

## F    SUPPLEMENTARY TO EXPERIMENTS SETUP

### F.1    STOCHASTIC LORENZ ATTRACTOR SETUP

Lorenz attractor is a system of three ordinary differential equations:

$$\frac{d\mathbf{x}}{dt} = \sigma(\mathbf{y} - \mathbf{x}), \quad \frac{d\mathbf{y}}{dt} = \mathbf{x}(\rho - \mathbf{z}) - \mathbf{y}, \quad \frac{d\mathbf{z}}{dt} = \mathbf{xy} - \beta\mathbf{z} , \tag{25}$$

where $\sigma$, $\rho$, and $\beta$ are system parameters. We set $\sigma = 10$, $\rho = 28$ and $\beta = 8/3$ to make the system chaotic. We simulate the trajectories by RK4 with a step size of 0.01. To make it stochastic, we add process noise to the transition, which is a mixture of two Gaussians $0.5\mathcal{N}(\mathbf{m}_0, \mathbf{P}) + 0.5\mathcal{N}(\mathbf{m}_2, \mathbf{P})$, where

$$\mathbf{m}_0 = \begin{bmatrix} 0 \\ 1 \\ 0 \end{bmatrix}, \quad \mathbf{m}_1 = \begin{bmatrix} 0 \\ -1 \\ 0 \end{bmatrix}, \quad \mathbf{P} = \begin{bmatrix} 0.06 & 0.03 & 0.01 \\ 0.03 & 0.03 & 0.03 \\ 0.01 & 0.03 & 0.05 \end{bmatrix} . \tag{26}$$

Besides, we add a Gaussian noise with zero mean and diagonal standard deviation $[0.6, 0.4, 0.8]$ as the observation noise. Totally, we simulate 5000 sequences as training set, 200 sequences as validation set, and 800 sequences as test set. For evaluation of Wasserstein distance, we simulate 10 groups of sequences additionally. Each group has 100 sequences with similar initial observations.

### F.2    TAXI TRAJECTORIES SETUP

The full dataset is very large and the length of trajectories varies. We select the trajectories inside the Porto city area with length in the range of 30 and 45, and only extract the first 30 coordinates of each trajectory. Thus we obtain a dataset with a fixed sequence length of 30. We split it into the training set of size 86386, the validation set of size 200, and the test set of size 10000.

### F.3 U.S. POLLUTION DATA SETUP

The U.S. pollution dataset consists of four pollutants (NO2, O3, SO2 and O3). Each of them has 3 major values (mean, max value, and air quality index). It is collected from counties in different states for every day from 2000 to 2016. Since the daily measurements are too noisy, we firstly compute the monthly average values of each measurement, and then extract non-overlapped segments with the length of 24 from the dataset. Totally we extract 1639 sequences as training set, 25 sequences as validation set, and 300 sequences as test set.

### F.4 NBA SPORTVU DATA SETUP

We use a sliding window of the width 30, and the stride 30 to cut the long sequences to short sequences of a fixed length 30. We split them into the training set of size 8324, the validation set of size 489, and the test set of size 980.

## G IMPLEMENTATION DETAILS

Here, we provide implementation details of VDM models used across the three datasets in the main paper. VDM consists of

- encoder: embed the first observation $\mathbf{x}_0$ to the latent space as the initial latent state $\mathbf{z}_0$.
- transition network: propagate the latent states $\mathbf{z}_t$.
- decoder: map the latent states $\mathbf{z}_t$ and the recurrent states $\mathbf{h}_t$ to observations $\mathbf{x}_t$.
- inference network: update the latent states $\mathbf{z}_t$ given observations $\mathbf{x}_t$.
- latent GRU: summarize the historic latent states $\mathbf{z}_{\leq t}$ in the recurrent states $\mathbf{h}_t$.
- discriminator: be used for adversarial training.

The optimizer is Adam with the learning rate of $1e - 3$. In all experiments, the networks have the same architectures but different sizes. The model size depends on observation dimension $\mathbf{d_x}$, latent state dimension $\mathbf{d_z}$, and recurrent state dimension $\mathbf{d_h}$. The number of samples used at each time step in the training is $2\mathbf{d_z} + 1$. If the model output is variance, we use the exponential of it to ensure its non-negative.

- Encoder: input size is $\mathbf{d_x}$; 3 linear layers of size 32, 32 and $2\mathbf{d_z}$, with 2 ReLUs.
- Transition network: input size is $\mathbf{d_h}$; 3 linear layers of size 64, 64, and $2\mathbf{d_z}$, with 3 ReLUs.
- Decoder: input size is $\mathbf{d_h} + \mathbf{d_z}$; 3 linear layers of size 32, 32 and $2\mathbf{d_x}$, with 2 ReLUs.
- Inference network: input size is $\mathbf{d_h} + \mathbf{d_x}$; 3 linear layers of size 64, 64, and $2\mathbf{d_z}$, with 3 ReLUs.
- Latent GRU: one layer GRU of input size $\mathbf{d_z}$ and hidden size $\mathbf{d_h}$
- Discriminator: one layer GRU of input size $\mathbf{d_x}$ and hidden size $\mathbf{d_h}$ to summarize the previous observations as the condition, and a stack of 3 linear layers of size 32, 32 and 1, with 2 ReLUs and one sigmoid as the output activation, whose input size is $\mathbf{d_h} + \mathbf{d_x}$.

**Stochastic Lorenz attractor.** Observation dimension $\mathbf{d_x}$ is 3, latent state dimension $\mathbf{d_z}$ is 6, and recurrent state dimension $\mathbf{d_h}$ is 32.

**Taxi trajectories.** Observation dimension $\mathbf{d_x}$ is 2, latent state dimension $\mathbf{d_z}$ is 6, and recurrent state dimension $\mathbf{d_h}$ is 32.

**U.S. pollution data**[6] Observation dimension $\mathbf{d_x}$ is 12, latent state dimension $\mathbf{d_z}$ is 8, and recurrent state dimension $\mathbf{d_h}$ is 48.

---

[6]https://www.kaggle.com/sogun3/uspollution

**NBA SportVu data.**  Observation dimension $\mathbf{d_x}$ is 2, latent state dimension $\mathbf{d_z}$ is 6, and recurrent state dimension $\mathbf{d_h}$ is 32.

Here, we give the number of parameters for each model in different experiments in Table 6.

Table 6: Number of parameters for each model in three experiments. VDM, AESMC, VRNN, and RKN have comparable number of parameters. CF-VAE has much more parameters.

|          | RKN   | VRNN  | CF-VAE  | AESMC | VDM   |
|----------|-------|-------|---------|-------|-------|
| Lorenz   | 23170 | 22506 | 7497468 | 22218 | 22218 |
| Taxi     | 23118 | 22248 | 7491123 | 22056 | 22056 |
| Pollution| 35774 | 33192 | 8162850 | 31464 | 31464 |
| SportVu  | 23118 | 22248 | 7491123 | 22056 | 22056 |

## H  ADDITIONAL EVALUATION RESULTS

We evaluate more variants of VDM in the chosen experiments to investigate the different choices of sampling methods (Monte Carlo method, and SCA) and weighting functions (Eqs. (27) and (28)). In addition to Eq. (27) described in the main text, we define one other choice in Eq. (28).

$$\omega_t^{(i)} = \omega(\mathbf{s}_{t-1}^{(i)}, \mathbf{x}_t)/k := \mathbb{1}(i = \arg\max_j p(\mathbf{x}_t \mid \mathbf{h}_{t-1} = \mathbf{s}_{t-1}^{(j)})) \qquad (27)$$

$$\omega_t^{(i)} = \omega(\mathbf{s}_{t-1}^{(i)}, \mathbf{x}_t)/k := \mathbb{1}(i = j \sim \mathrm{Cat}(\cdot \mid \omega^1, \ldots, \omega^k)), \quad \omega^j \propto p(\mathbf{x}_t \mid \mathbf{h}_{t-1} = \mathbf{s}_{t-1}^{(j)}), \qquad (28)$$

We define the weighting function as an indicator function, in Eq. (27) we set the non-zero component by selecting the sample that achieves the highest likelihood, and in Eq. (28) the non-zero index is sampled from a categorical distribution with probabilities proportional to the likelihood. The first choice (Eq. (27)) is named with $\delta$-function, and the second choice (Eq. (28)) is named with categorical distribution. Besides, in VDM-Net, we evaluate the performance of replacing the closed-

Table 7: Definition of VDM variants

|                   | VDM($k=1$)  | VDM-MC+$\delta$ | VDM-SCA+Cat              | VDM-SCA+$\delta$ |
|-------------------|-------------|-----------------|--------------------------|------------------|
| Sampling method   | Monte-Carlo | Monte-Carlo     | SCA                      | SCA              |
| Weighting function| n.a.        | $\delta$-function | Categorical distribution | $\delta$-function |

form inference of the weighting function with an additional inference network. In Table 7, we show the choices in different variants. All models are trained with $\mathcal{L}_{\mathrm{ELBO}}\&\mathcal{L}_{pred}$.

### H.1  STOCHASTIC LORENZ ATTRACTOR

Table 8: Ablation study of VDM's variants on stochastic Lorenz attractor for three distance metrics (see main text). The variants are defined in Table 7. All variants give comparable quantitative results.

|             | VDM($k=1$)   | VDM-Net      | VDM-MC+$\delta$ | VDM-SCA+Cat   | VDM-SCA+$\delta$ |
|-------------|--------------|--------------|-----------------|---------------|------------------|
| Multi-steps | 25.03±0.28   | 26.65±0.15   | 24.67±0.16      | 24.69±0.16    | 24.49±0.16       |
| One-step    | -1.81        | -1.71        | -1.84           | -1.83         | -1.81            |
| W-distance  | 7.31±0.002   | 7.68±0.002   | 7.31±0.005      | 7.30±0.009    | 7.29±0.003       |

### H.2  TAXI TRAJECTORIES

### H.3  U.S. POLLUTION DATA

Table 9: Ablation study of VDM's variants on taxi trajectories for three distance metrics (see main text). The variants are defined in Table 7. VDM-SCA+$\delta$ outperforms other variants and approaches our default VDM (trained with $\mathcal{L}_{adv}$ additionally).

|  | VDM($k=1$) | VDM-Net | VDM-MC+$\delta$ | VDM-SCA+Cat | VDM-SCA+$\delta$ |
|---|---|---|---|---|---|
| Multi-steps | 3.26±0.001 | 3.68±0.002 | 3.17±0.001 | 3.09±0.001 | 2.88±0.002 |
| One-step | -2.99 | -2.74 | -3.21 | -3.24 | -3.68 |
| W-distance | 0.69±0.0005 | 0.79±0.0003 | 0.70±0.0008 | 0.64±0.0005 | 0.59±0.0008 |

Table 10: Ablation study of VDM's variants on U.S. pollution data for two distance metrics (see main text). The variants are defined in Table 7. VDM-SCA+$\delta$ outperforms other variants.

|  | VDM($k=1$) | VDM-Net | VDM-MC+$\delta$ | VDM-SCA+Cat | VDM-SCA+$\delta$ |
|---|---|---|---|---|---|
| Multi-steps | 42.33±0.11 | 52.44±0.04 | 40.33±0.03 | 39.58±0.09 | 37.64±0.07 |
| One-step | 7.97 | 10.70 | 8.12 | 7.82 | 6.91 |

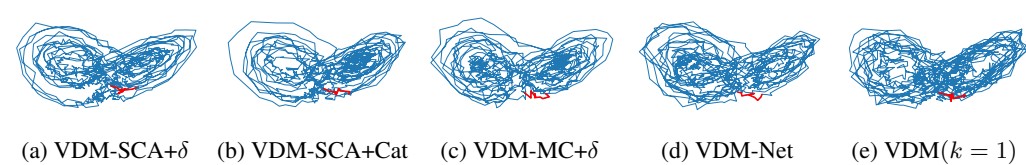

(a) VDM-SCA+$\delta$  (b) VDM-SCA+Cat  (c) VDM-MC+$\delta$  (d) VDM-Net  (e) VDM($k=1$)

Figure 7: Generated trajectories of stochastic Lorenz attractor from VDM variants. The first ten observations (red) are obtained from models given the first 10 true observations. The rest 990 observations (blue) are predicted. We can see, all variants give very good qualitative results. Since the fundamental dynamics is govern by ordinary differential equations, the transition at each time step is not highly multi-modal. Once the model is equipped with a stochastic transition, it is able to model this dynamics.

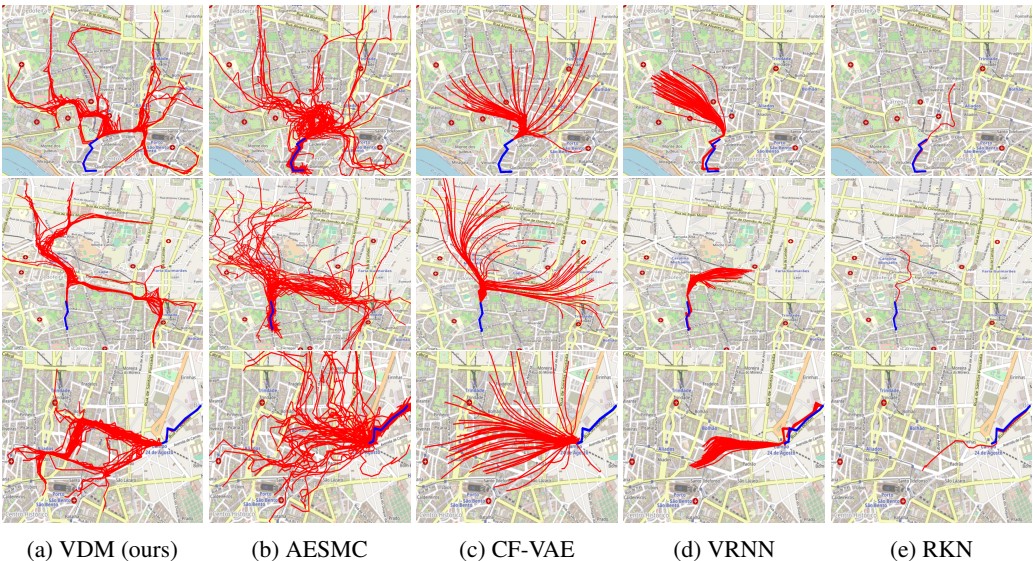

(a) VDM (ours)  (b) AESMC  (c) CF-VAE  (d) VRNN  (e) RKN

Figure 8: Generated 50 taxi trajectories in 3 different areas from VDM and the baselines. All models are required to predict the future continuations (red), based the beginning of a trajectory (blue). VDM generates more plausible trajectories compared with the baselines. While the generated trajectories from VDM follow the street map, the generated trajectories from all baselines are physically impossible. AESMC and CF-VAE can capture the general evolving direction, but suffer from capturing the multi-modality at each time step.

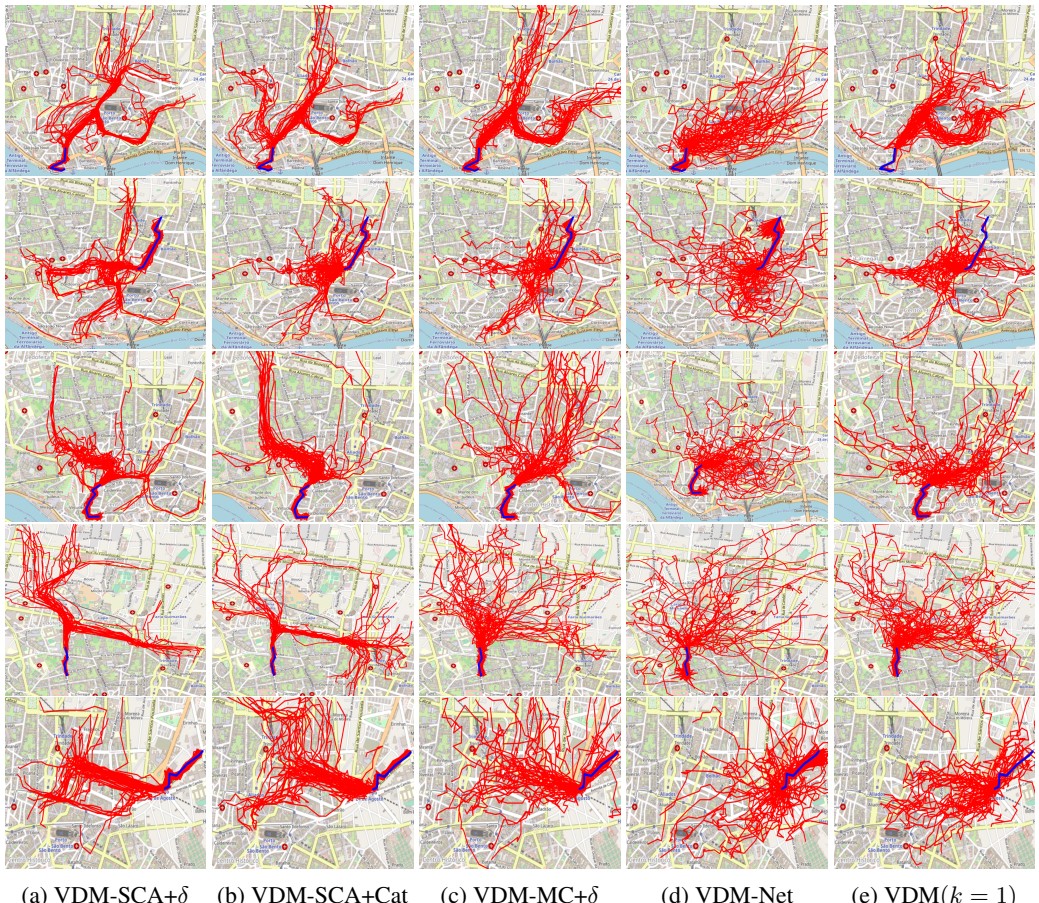

(a) VDM-SCA+$\delta$  (b) VDM-SCA+Cat  (c) VDM-MC+$\delta$  (d) VDM-Net  (e) VDM($k = 1$)

Figure 9: Generated 50 taxi trajectories from VDM variants. All models are required to predict the future continuations (red), based the beginning of a trajectory (blue). VDM-SCA+$\delta$ achieves the best qualitative results among all variants. VDM-SCA+$\delta$ can generate plausible trajectories, even it is trained without the adversarial term $\mathcal{L}_{adv}$. We can see, for the weighting function, Eq. (27) is better than Eq. (28), and for the sampling method, SCA is better than Monte-Carlo method.

