# OpenReview forum: "Variational Dynamic Mixtures"
_ICLR.cc/2021/Conference — Reject_

### Official Review · AnonReviewer1 · 2020-10-25
**A deep probabilistic model for time series forecasting is proposed.  Detailed description for mathematics is required.**

**Rating:** 4
**Confidence:** 3

**Review:**

This paper presented the variational dynamic mixtures as a deep probabilistic model for time series forecasting. The research issue, called the taxi trajectory prediction problem, is addressed. Some comments are provided.

Pros:
A new solution to mixture density network as a kind of generative model with latent states and multinomial observations was proposed. The detailed experiments were addressed. New evaluation metric was introduced.

Cons:
There are a number of notations and variables which were not clearly defined. This matter made the reading to be easily confused. A clear algorithm or working flow for complicated system was missing. Some descriptions were not clear.

---

> ### Author Response · Authors · 2020-11-19
> **We added an algortihm and revised section 3 ("Inference Model").**
>
> Thank you for taking the time to review our paper. We are glad that you appreciate the novelty of VDM as well as the extensive empirical study. As pointed out above, we added an additional experiment that further strengthened our claims.
> * We appreciated your idea of an algorithmic overview of our method. We included an algorithm for the generative model and the inference model in Appendix C.
> * We acknowledge the fact that the presentation of VDM could be further improved and revised section 3 ("Inference Model") accordingly. We tried to address our comment as good as we could in terms of specifying all distributions and variables. Please let us know if you have any further suggestions, and we are happy to adapt them.

---

### Official Review · AnonReviewer2 · 2020-10-26
**New inference model for capturing multi-modal dynamics**

**Rating:** 7
**Confidence:** 3

**Review:**

Summary

This paper introduces variational dynamic mixtures (VDM), a new variational family, and demonstrates that using VDM to model the approximate posterior in sequential latent variable models can better capture multi-modality in data. VDM includes a distribution over recurrent states in the inference model, such that a sampling-based marginalization of this distribution reduces the approximate posterior to a mixture model. Setting the weights such that only the most probable mixture component is selected allows other mixture components to capture other modes. The authors validate VDM on both synthetic and real multimodal datasets, which outperform baselines with respect to negative log-likelihood and a new empirical Wasserstein distance.

Positives

+ This paper tackles an important and well-motivated problem: capturing multi-modality in data. This is very practical and I believe will be of interest to the ICLR community.

+ Paper is well-written and easy to follow. I appreciate that the authors highlight their design decisions in the main text, while also providing alternatives and/or ablation studies in the appendix.

+ VDM improves performance while also using a non-autoregressive generative model, compared to baselines with more powerful generative models (e.g. VRNN). This highlights the effectiveness of their inference model, also illustrated in the synthetic experiment in Figure 3. The inference model in VDM is also quite general, as a single-sample approximation in their inference model is equivalent to the inference model in VRNN.

Concerns

- I think this paper would benefit from one additional dataset where the multimodality is inherent in the data. Taxi trajectories are multimodal but also highly structured (trajectories must be on roads), so a different dataset you can consider can be pedestrian or sport trajectory datasets, where the data is also inherently multimodal but also less structured. I think some baseline models can do better in this setting (at least qualitatively), so I’m curious if VDM still convincingly outperforms them. My main concern is that the pollution data is synthetically multimodal (because I think that with contextual information the data is more periodic) and the Lorenz attractor experiment only highlights that VDM can handle stochasticity, which is also present in trajectory data.

- The related work is missing a section about other methods that try to capture multimodality. For instance, VRNNs are known to not capture multimodality well, and there have been extensions along this direction such as in [1]. There’s also another line of work that introduces mutual information between trajectories and latent variables in the objective, such as in [2]. The “sequence forecasting” paragraph can be omitted/combined with “neural recurrent models”.

- The results on the taxi dataset look good. It would be great if you can also provide analysis on the resulting latent space, similar to what was done in Figure 3.

[1] Goyal et al. Z-Forcing: Training Stochastic Recurrent Networks

[2] Li et al. InfoGAIL: Interpretable Imitation Learning from Visual Demonstrations

Minor Comments

- The bold/thin lines in Figure 2 can be hard to distinguish. I recommend using a dotted line instead.

- In Figure 3, how many timesteps are there in total? Are the blue/orange trajectories in the left plots corresponding to the blue/orange clusters in the middle/right plots?

- Tables 1,2 and 3 all cut through paragraphs in the middle, which can be distracting.

- Some quotations on page 5 use close quotations ” on both sides.

---------

Post-rebuttal comments

Thank you for adding the additional experiments and analysis. The results with the basketball dataset (Table 4, Figure 6) and the visualization of the latent distribution (Figure 5) address my inital concerns and showcase the versatility of VDM. I've increased my score from 6 to 7.

---

> ### Author Response · Authors · 2020-11-19
> **We added a new experiment, a new visualization, and discussed the related work.**
>
> We greatly appreciate your feedback, including your suggestion to try VDM on another dataset with less restrictive dynamics.
> * We ran additional experiments that showed that VDM also outperforms competing approaches in forecasting trajectories of basketball players. The players can move anywhere on the court and are not constrained by the street map in the same way the taxi trajectories are, which shows that the approach is quite versatile. We included the new experiments in the updated write-up (see Table 4 and Figure 6).
> * We also adopted your suggestion to visualize the latent distributions of VDM on the taxi trajectories. In Figure 5, you can find a trajectory on the map (left) with three waypoints marked yellow, blue, and purple. These are points where trajectories can split. On the right-hand side of Fig. 5, we visualize the predictive prior at these locations, comparing three different inference methods. Among all baselines under consideration, VDM is the only approach with a multi-modal predictive distribution.
> * We are also grateful for the additional references you provided (we have added them), and we also took your other comments into account.

---

### Official Review · AnonReviewer4 · 2020-10-28
**Good paper on interesting topic, some clarifications needed.**

**Rating:** 7
**Confidence:** 4

**Review:**

**Variational dynamic mixtures**

The paper extends the VRNN to cater for multi-modality in the probability distribution governing a dynamic system. This is particularly important when average trajectories are highly unlikely or even physically impossible. To achieve this, the authors start from VRNN and alter the inference model so that it uses stochastic recurrent states and a mixture variational posterior distribution (with 0/1 weights to trigger only the most likely mixture components encouraging multi-modality).
As a minor contribution they propose a new evaluation metric for measuring the diversity of generations based on Wasserstein distance. This is important in their case when the likelihood evaluation may favour generations from a single mode - a situation their model shall prevent.

The paper is very well motivated (with taxi trajectory prediction as a running use case) and well positioned with respect to the state of the art.
The experimental evaluation is convincing, using both synthetic and real experiments to support the claims and shows advantages over baselines. Ablation studies examine the importance of some more ad-hoc choices (showing these help but are not critical).
The paper is well written and structured to help the reader follow the main thoughts.
However, there are some points in the mathematical formulation of the model which raise questions and deserve to be explained better - see below.

For this reason I recommend not accepting the paper for now but I'm am very much willing to improve my score significantly once these will have been clarified.

* From Fig2 it seems that the stochastic states s_t do not exist in the generative model, they only live in the inference model. Right?

* From eq. (1) and Fig2a we have $h_{t} = \phi^{GRU}(z_{t}, h_{t-1})$ and the generative distribution of $p(z_{t+1} | z_{\leq t})$ is a function $\phi^{tra}(h_{t})$. Is it correct to think about this as the prior distribution for $z_{t+1}$?

* My understanding of eq. (3) is that we have $\widetilde{q}(s_{t-1} | x_{<t}) = 1$ if $s_{t-1} = h_{t-1}$ and zero otherwise. Is that right?

* You say $q(s_{t-1} | x_{\leq t}) = \omega(s_{t-1}, x_t) \, \widetilde{q}(s_{t-1} | x_{<t})$.
    * How can you condition $s_{t-1}$ on the future value $x_t$? (Also in comparison to equation (4), where you assume $q(z_t | x_{\leq T}) = q(z_t | x_{\leq t})$ for all $t$ thus avoiding dependence on future values of $x$.
    * If my understanding of eq (3) is correct (see above), this is trivially 1 or 0 irrespective of $\omega$.

* In eq (4) you say $q(z_{1:T} | x_{1:T}) = \Pi_{t=1}^T q(z_{t} | x_{\leq t})$.  Is this correct? I would expect $\Pi_{t=1}^T q(z_{t} | z_{\lt t}, x_{\leq t})$.

* Again in eq (4) you say the final result is $\Pi_{t=1}^T \int q(z_{t} | s_{t-1}, x_{t}) q(s_{t-1} | x_{\leq t}) d s_{t-1}$. Is this correct? I would expect $\Pi_{t=1}^T \int q(z_{t} | s_{t-1}, \mathbf{x_{\leq t}}) q(s_{t-1} | x_{\leq t}) d s_{t-1}$. Does this mean you assume the conditional independence $q(z_{t} | s_{t-1}, \mathbf{x_{\leq t}}) = q(z_{t} | s_{t-1}, \mathbf{x_{t}})$?

* samples $s_{t-1}^{(i)}$ in eq (5) are constructed by sampling $z_{t-1}^{(i)} \sim q(z_{t-1} | x_{<t})$ and passing it through the recurrent net $s_{t-1}^{(i)} \gets h_{t-1}^{(i)} = \phi^{GRU}(z_{t-1}^{(i)}, h_{t-2})$, right?

* eq (5) and annex A: How come $\omega$ is a function of $x_t$ only and not of $x_{<t}$. Given your importance sampling argument, $q$ conditions on $x_{<t}$ as well. Or do you assume the conditional indpepence $q(x_t | s_{t-1}, x_{<t}) = q(x_t | s_{t-1})$?

* what is $p(x_t | s_{t-1})$ used in eq (6)? This looks like a generative distribution ($p$) but has not been defined before and $s_{t-1}$ do not exist in the generative model (figure 1a).

* ELBO proof in annex B: can you please provide details (equations, detailing also the conditional independence assumptions you take) for getting from eq (7) to eq (8)?

* Wasserstein distance in eq (14) - the differences are calculated between the generated trajectories and the corresponding true sample (the *group sample*) or all the n true samples?

AFTER REVIEW UPDATE: I find the revised version much improved explaining the inference model much more clearly. The lack of clarity was for me the main reason for evaluating the paper as below the acceptance threshold despite the fact that otherwise I found the paper to be good and useful for the community. As the lack of clarity has now been, in my view, resolved, I increase my score to 7 - Good paper, accept.

---

> ### Author Response · Authors · 2020-11-19
> **We appreciated your detailed feedback and revised section 3 accordingly.**
>
> We highly appreciated your detailed feedback, which immensely helped us revise the paper. We revised section 3 accordingly (focusing on the inference procedure), where we tried to address several of your comments and questions. In particular, we better motivated the role of the recurrent stats s_t. As we explain there, the stochastic latent state allows passing more information about the distribution of past latent states through the (forward) transition model. We also updated the text to be more explicit about our variational distribution's factorization and conditional independence assumptions.
>
> Below, we answer your clarification questions (we enumerated the questions 1-11)
>
> **Q1**: ''Does $s_t$ only live in the inference model?''
>
> **A**: Yes, only the inference model is augmented with the recurrent state $s_t$, by marginalizing it out again we get a distribution over $z_t$.
>
> **Q2**: ''Is it correct to think about $p(z_{t+1}|z_{\leq t})$ as the prior distribution for $z_{t+1}$?''
>
> **A**: Yes, you can think of  $p(z_{t+1}|z_{\leq t})$ as a prior for $z_{t+1}$.
>
> **Q3**: ''Is $\tilde{q}(s_{t-1}|x_{<t}) =1$, if $s_{t-1}=h_{t-1}$ and zero otherwise?''
>
> **A**: Note that here $z_{t-1}$ is not a single sample but a random variable. If we approximate the integration by a single sample form $q(z_{t-1} | x_{<t})$ then yes, you are right. If we use multiple ($k$) samples, it equals $1/k$ when $s_{t-1}=h_{t-1}$, otherwise zero. To avoid confusion we have moved Eqn (3) to a footnote and now only describe how to sample from $\tilde{q}(s_{t-1}|x_{<t})$ in Eqn (4).
>
> **Q4**: ''$q(s_{t-1}|x_{\leq t})= \omega(s_{t-1}, x_{t})\tilde{q}(s_{t-1}|x_{<t})$''
>
> - ''How can $s_{t-1}$ condition on $x_{t}$?''
>
> - **A**: That's a great nuance of our method you picked up on. The factor $q(z_t | x_{\leq t})$ on $z_t$ conditions only on data from the past and present, but in Eqn (4) we can see that our marginal consistency assumption marginalizes over the distribution of the *last* hidden state $s_{t-1}$ conditioning on past observations AND the *current* observation $x_t$. Mathematically, there is no contradiction in using a distribution over $s_t-1$ that looks into the present and uses $x_t$. There was an arrow missing in Fig 2b. We have now added an arrow for the dependence of $s_{t-1}$ on $x_{t}$.
>
> - ''Is $q(s_{t-1}|x_{\leq t})$ trivially 1 or 0?''
>
> - **A**: No, Since $\tilde{q}(s_{t-1}|x_{<t})$ is not trivially 1 or 0 (see Q3). This factorization is meaningful.
>
> **Q5**: ''Why is not $q(z_{1:T}|x_{1:T}) =\prod_1^T q(z_t | z_{<t}, x_{\leq t})$?''
>
> **A**: To answer your question, let’s first consider the difference between our work and VRNN [1] in terms of posterior factorization. $q(z_{1:T}|x_{1:T}) =\prod_1^T q(z_t | z_{<t}, x_{\leq t})$ corresponds to a structured posterior factorization assumption as assumed in the VRNN. We instead assume a mean field factorization (part 1 of Eqn (4)) with temporal structure coming from the marginal consistency assumption (part 2 of Eqn (4)).  By marginalizing over $s_{t-1}$, $z_t$ no longer conditions on past states. Now only the variational parameters of $q(z_t | x_{\leq t})$ (which are computed by performing the marginalization) contain information about past states (e.g. $\mu_{z,t}^{(i)}$ is a function of $z_{t-1}^{(i)}$, and $h_{t-2}$ ).
>
> **Q6**: ''Do you assume the conditional independence assumption $q(z_t|x_{\leq t}, s_{t-1}) =q(z_t|x_t,s_{t-1})$?''
>
> **A**: Yes, we make the conditional independence assumption that $q(z_t|x_{\leq t}, s_{t-1}) =q(z_t|x_t,s_{t-1})$ depends only on the current $x_t$. All dependencies on previous observations $x_{<t}$ and states $z_{<t}$ are managed through marginalizing over $s_{t-1}$. One reason we chose this conditional independence assumption here, is that only the current observation (fixed size) is used together with $s_{t-1}$ as input to the inference network $\phi^{inf}$. Also, $s_{t-1}$ already contains information of past observations and states. The inference network learns how to combine $s_{t-1}$ with the new observations such that convolution with $q(z_t|x_t, s_{t-1})$ turns the distribution over $s_{t-1}$ into a distribution over $z_t$.
>
> **Q7**: ''How to take samples $s_{t-1}^{(i)}$?''
>
> **A**: Yes, you are right. We now explicitly express this in Eqn (4).
>
> References
>
> [1] Chung, Junyoung, et al. "A recurrent latent variable model for sequential data." Advances in neural information processing systems. 2015.

---

> > ### Author Response · Authors · 2020-11-19
> > **Continue the answering.**
> >
> > **Q8**: ''How come $\omega$ is a function of $x_{t}$ only and not of $x_{<t}$?''
> >
> > **A**: Above Eqn (4), we specify the factorization assumption $q(s_{t-1}|x_{\leq t})= \omega(s_{t-1}, x_{t})\tilde{q}(s_{t-1}|x_{<t})$. A more general choice would have been to make $\omega$ a function of $x_{<t}$ as well, but we make a stronger conditional independence assumption which we are now more explicit about in the text.
> >
> > In appendix A, Eqn (15) should involve $q(x_t|s_{t-1}, x_{<t})$. Without mentioning it explicitly we used the assumption $q(x_t | s_{t-1}, x_{<t}) = p(x_t | h_{t-1} = s_{t-1}, x_{<t}) = p(x_t | h_{t-1} = s_{t-1})$. In the generative model, conditioning on a specific value for $h_{t-1}$ drops the dependence on previous $x_{<t}$.
> >
> > **Q9**: ''what is $p(x_t|s^{(i)}_{t-1})$ used in Eqn (6)?''
> >
> > **A**: Thanks for noticing the typo. $s_t$ doesn’t appear in the generative process and the expression in Eqn. (6) should be $p(x_t|h_{t-1} = s^{(i)}_{t-1})$.
> >
> > **Q10**: ''More details in ELBO proof in appendix B.''
> >
> > **A**: We added missing details to our ELBO derivation.
> >
> > **Q11**: ''The calculation of the Wasserstein distance.''
> >
> > **A**: The Wasserstein distance is calculated by taking the distance between the generated trajectories and the corresponding true sample (the “*group sample*”).

---

> > ### Comment · AnonReviewer4 · 2020-11-23
> > **Clarifications in text help a lot**
> >
> > Thank you for all your responses and clarifications, I find the new version of the paper (the inference model) much improved.

---

### Author Response · Authors · 2020-11-24
**summary of changes**

We thank all reviewers for their helpful feedback on our paper, in which we have presented VDM, a sequential latent variable model with a novel variational inference method that is particularly suited for multi-modal dynamics.

Here, we would like to summarize our changes:

* Thanks to a suggestion by R2, we added experiments on a basketball dataset that has less restrictive dynamics than the taxi data (see Table 4 and Fig. 6 in the updated draft).

* We also adopted a suggestion by R2 to visualize the latent distributions of VDM on the taxi trajectories (see Fig. 5).

* Finally, we extensively edited section 3 on the inference model based on feedback from R4, who already considered the new version to be a significant improvement.

We hope that these improvements will let the reviewers increase their scores.

---

### Decision · Program_Chairs · 2021-01-07
**Final Decision**

**Decision:**

Reject

**Comment:**

The paper proposes a variational family of distributions for posterior estimation in sequential latent variable models. The paper does so by extending variational recurrent neural networks so as to use a variational-mixture posterior and capture more realistic multi-modalities in the data.

During the review process, it was suggested to improve the clarity of the paper, provide results on an additional dataset and a visualization of the latent distributions. I commend the authors for addressing these issues satisfactorily.

Overall, the paper is well-motivated and well-written. However, when considering the novelty of the paper, although none of the reviewers raised this issue, I believe the paper heavily relies on previously proposed ideas and therefore, its contribution can be seen as incremental. Additionally, something important to highlight is in section 2, with regards to deep state-space models (SSMs). The authors make a rather strong claim with regards to assumptions on the variational distribution. However, one can find out-of-the-box implementations where this is not the case, see e.g. https://pyro.ai/examples/dmm.html that implements deep Markov models with posteriors based on inverse autoregressive flows. A comparison with such approaches may be also required.